# Ligands of Adrenergic Receptors: A Structural Point of View

**DOI:** 10.3390/biom11070936

**Published:** 2021-06-24

**Authors:** Yiran Wu, Liting Zeng, Suwen Zhao

**Affiliations:** 1iHuman Institute, ShanghaiTech University, Shanghai 201210, China; wuyr@shanghaitech.edu.cn (Y.W.); zenglt@shanghaitech.edu.cn (L.Z.); 2School of Life Science and Technology, ShanghaiTech University, Shanghai 201210, China

**Keywords:** adrenergic receptor, adrenoreceptor, aminergic receptor, GPCR, activation mechanism, receptor–ligand interactions, selectivity, allosteric modulator, structure-based drug design

## Abstract

Adrenergic receptors are G protein-coupled receptors for epinephrine and norepinephrine. They are targets of many drugs for various conditions, including treatment of hypertension, hypotension, and asthma. Adrenergic receptors are intensively studied in structural biology, displayed for binding poses of different types of ligands. Here, we summarized molecular mechanisms of ligand recognition and receptor activation exhibited by structure. We also reviewed recent advances in structure-based ligand discovery against adrenergic receptors.

## 1. Introduction

Adrenergic receptors (adrenoreceptors, ARs) are membrane proteins mediating the actions of epinephrine and norepinephrine. Epinephrine and norepinephrine (also called (–)-adrenaline and (–)-noradrenaline, respectively) are neurotransmitters of the sympathetic nervous system and the central nervous system, and function as hormones secreted by the adrenal medulla. There are nine members of adrenergic receptors in humans (Figure 1a), widely distributed throughout the body and playing key roles in many important physiological processes, such as response to stress, control of heart rate and blood pressure, and regulation of metabolism [1,2,3]. Besides epinephrine and norepinephrine as emergency medicines, adrenergic receptors are also the target of many medications in treatment of various conditions, including hypertension, hypotension, heart failure, arrhythmias, and asthma [1,4]. In the 21st century, the development of new drugs against adrenergic receptors has slowed down, but the once most-neglected member β_3_-adrenergic receptor is raising more attention than ever (Figure 1b).

The adrenergic receptors belong to the rhodopsin family/class A G protein-coupled receptors (GPCRs). The β_2_-adrenergic receptor (β_2_AR) is a prototype GPCR for studies of sequence, structure, and function: β_2_AR is the first GPCR cloned [5], the second GPCR was determined for atomic structure [6,7] (only after rhodopsin), and the first GPCR solved for the active state structure in a functional complex [8] (with G_s_ protein heterotrimer). β-arrestins, the cytoplasmic partners responsible for desensitization of most GPCRs, were named after the β_2_AR [9,10]. Turkey β_1_AR has been comprehensively studied in structural biology (in this article, all structure of β_1_AR refers to turkey β_1_AR unless noted). A complex of β_1_AR and β-arrestin-1 was reported in 2020 [11].

β_2_AR and β_1_AR are among the GPCRs with the largest numbers of structures obtained, providing rich information for ligand recognition and activation mechanisms (β_2_AR has 38 structures for 19 ligands and β_1_AR has 30 structures for 17 ligands). In 2019, structures of the three α_2_ adrenergic receptors were also reported, disclosing the structural basis for the receptor-type selectivity of ligands. In this review, we summarized current achievements of structural study in adrenergic receptors and advances in drug discovery based on these structures.

## 2. Adrenergic Receptors in the Aminergic Receptor Subfamily

The nine adrenergic receptors in human body can be classified into three major types: α_1_, α_2_, and β, each having three subtypes. Members within a major type are highly similar in sequence and function, but in different major types are distinct in these aspects (see next section for detail). All the adrenergic receptors belong to the aminergic receptor subfamily, which has 42 members, forming a single clade in the phylogenetic tree of GPCRs [18,19,20,21]. The aminergic receptors bind monoamine neurotransmitters, acetylcholine, or trace amines, and share common features in sequence, structure, and function (Figure 2): in the orthosteric site (binding site of endogenous ligand), a conserved D3.32 (Ballesteros–Weinstein numbering, BWN [22]) forms a salt bridge to the positively charged amino group in ligand, and Y7.43 and W7.40 stabilize this salt bridge through a hydrogen bond and π-π interactions (Figure 3b). Because of the shared ligand recognition mechanism, polypharmacology (drugs interacting with multiple targets) is common in aminergic receptors. For example, ergotamine (an alkaloid produced by fungus) can bind to 22 aminergic receptors with *K*i less than 1 μM [23]. Other notable cases include clozapine [24] and apomorphine [25]. Therefore, selectivity is a major concern in drug discovery against aminergic receptors.

The endogenous ligands of adrenergic receptors, epinephrine and norepinephrine, are monoamine neurotransmitters. Monoamine (also include dopamine, serotonin, and histamine) contains an amino group connected to an aromatic ring by a two-carbon chain (Figure 2b). When binding to receptors, the aromatic ring forms π-π interactions to F6.51 and F6.52 (Figure 2a and Figure 3b). Among the monoamines, dopamine is most similar to epinephrine and norepinephrine in chemical structure: each contains a catechol (1,2-dihydroxybenzene) group (thus collectively referred to as catecholamines). The catecholamines are products of three successive steps in the tyrosine metabolism pathway. Their receptors recognize the catechol hydroxy groups by S5.42 and S5.46 (Figure 2a and Figure 3b), two serine residues in TM5 that appear in all members of adrenergic and dopamine receptors. These two positions are non-conserved in other aminergic receptors. Among the catecholamines, ligand–receptor recognition is not strictly specific: dopamine also acts on adrenergic receptors [4,26,27], while epinephrine and norepinephrine are reported to stimulate some dopamine receptor subtypes [28,29].

## 3. Major Types and Subtypes of Adrenergic Receptors

Adrenergic receptors are among the most studied GPCRs. Research on their roles in sympathetic nerve stimulation started at the beginning of the 20th century. In the pre-molecular biology era, major types and subtypes of adrenergic receptors were identified through their pharmacological characteristics and tissue locations. All the nine members were cloned by the early 1990s, before the human genome project was completed [1,32].

Major types largely correspond to the sequences and functions of adrenergic receptors. Members within a major type share 51–64% sequence identities at full-length, while in different major types, the values decrease to 30–40% (Figure 3a). This is the basis: adrenergic receptors in each major type have similar affinities for many ligands and mainly couple to a specific G protein type (Table 1). Notably, pocket residues of α_1_ARs and α_2_ARs are more similar, thus they share some ligands that do not target βARs. For the distinct cell and tissue responses of adrenergic receptors, medications targeting different major types have different indications (Table 1). Two residues at the ligand binding site, 6.55 [33] and 7.39 [34], are reported to affect the major type selectivity of ligands (Figure 2a and Figure 3b,c).

Even within a major type, adrenergic receptor subtypes may have distinct effects, due to their tissue distribution. This is exhibited in the β adrenergic receptors: β_1_AR is mainly expressed in heart and adipose tissue, thus its activation produces cardiac stimulation and increased lipolysis; β_2_AR is ubiquitously expressed but with high levels in the smooth muscle, especially of the airway, thus its main effect is bronchodilation; and β_3_AR is mainly expressed in adipose tissue, thus is responsible for increased lipolysis [35]. Therefore, while using β antagonists to treat cardiovascular diseases, β_1_-selective agents have less asthma adverse effects than non-selective agents [36]. β_3_-selective agonists are widely investigated as anti-obesity agents, though they still face many challenges [37]. The benefits of subtype-selective drugs are also found in the other two major types: α_1A_/α_1D_ selective antagonist tamsulosin has less impact on blood pressure while treating benign prostatic hyperplasia [38]; and α_2C_ selective antagonists are in development as potential treatments for multiple psychiatric diseases [39]. Positions 5.39 [40], 5.43 [41], 6.58 [42], 6.62 [43] (ECL3), 7.32 [42], and 7.35 [44,45] are reported to affect the subtype selectivity of ligands. These residues are at the extracellular edge of the pocket (Figure 2a and Figure 3b,c).

## 4. Mechanisms of β Adrenergic Receptors

### 4.1. Activation Mechanism

For a class A GPCR, the most significant conformational changes during activation are of the outward movement of the cytoplasmic end of TM6 and rearrangement of TM7 (Figure 4a). About 90 receptors [52] share a common mechanism: the conformational changes are mediated by the microswitch of the P5.50-I3.40-F6.44 motif, induced by an agonist. How agonists trigger the microswitch of the P-I-F motif largely varies in different receptor families. For β_2_AR or β_1_AR, an agonist forms a hydrogen bond to S5.46, thus pulling TM5 closer to TM3/6. An antagonist, in contrast, lacks this hydrogen bond and tends to occupy the space with hydrophobic moiety, thus preventing the movement of TM5 (Figure 4b). This mechanism is presented in all β_2_/β_1_ agonists and antagonists (Figure 4c, representative ligands are listed in Appendix A).

Binding of the orthosteric agonist and coupling of G protein have an effect on each other: agonists have higher affinities in β_2_AR with the presence of G protein than in the absence of G protein [53]. Structure studies of agonists binding in different states of β_1_AR disclosed the molecular basis: active state structures have smaller pockets when comparing to agonist-bound inactive state structures (Figure 4d) [54]. Inferred from the mechanism, antagonists tend to fit in more expanded orthosteric pockets. This was established in different types of class A GPCRs, including lipid receptors [55,56,57,58], peptide receptors [59,60,61], and nucleotide receptors [62,63,64,65].

### 4.2. Biased Signaling

Some agonists stimulate the β-arrestin signaling with higher or lower EC_50_ comparing to the G protein signaling (called biased signaling). For βARs, separately targeting β-arrestin pathways or G protein pathways is considered a therapeutic potential for diseases, including heart failure and asthma [78,79]. Structures of GPCR in complex with arrestin have been obtained in several receptors [80,81,82,83] including β_1_AR [11]. Compared to G_s_-coupled β_1_AR, arrestin-coupled β_1_AR adopts a smaller degree of outward movement of TM6 and additional twist of TM7 (Figure 5a). Correspondingly, β_1_AR forms interactions to arrestin with more residues in TM2/7 and fewer residues in TM5/6 [11,66]. In accordance with this, NMR experiments of ^19^F-labeling on the cytoplasmic ends of TM6 and TM7 showed that arrestin-biased agonists induce more conformational change of TM7 than of TM6 for β_2_AR in solution [84]. Structures showed that arrestin-biased agonists carvedilol [85], bucindolo [86], formoterol [87], and BI-167107 [88] have an aryl-substituted alkyl tail attached to the amino N atom, and the bulky aromatic rings occupy an extended pocket formed by TM2/3/7 and ECL2 in β_2_AR or β_1_AR [8,11,41,66,75,89,90] (Figure 5). In contrast, G_s_-biased agonist salmeterol occupies this space but forms distinct interactions with a long aryloxyalkyl tail [70]. However, there are also mysteries of arrestin-biased signaling: an approved β agonist drug, isoetharine, is arrestin-biased in β_2_AR [91] but G_s_-biased in β_1_AR [92]. The structural mechanism remains to be explored.

### 4.3. Partial Agonism

Many βARs ligands in clinical use are partial agonists, which do not produce a full response, even at high concentration. NMR [84,93,94], single-molecule fluorescence resonance energy transfer [95], and double electron–electron resonance [94,96] results showed that partial agonists induce a smaller shift of conformational equilibrium of TM6, compared to full agonists. A well-identified key interaction affecting ligand efficacy (ratio of maximal response of the ligand to full response) is a hydrogen bond to N6.55 by a full agonist, but not a partial agonist (Figure 5b). Mutations of N6.55 and S5.43 (which forms hydrogen bond to N6.55) in β_2_AR reduce the efficacy of full agonist isoproterenol, strongly supporting that this hydrogen bond network is crucial to the full activation of βARs [70]. All full agonists can form a hydrogen bond to N6.55, either by 3-hydroxy in catechol (like epinephrine [31,43] or isoproterenol [54,66]) or groups mimicking catechol (such as formoterol [11] or BI-167107 [8,43,90]). Most partial agonists cannot form this hydrogen bond for replacement of the 3-hydroxy, including salmeterol [70], salbutamol [54,69], xamoterol [54], and clenbuterol. However, this cannot explain dobutamine, a partial agonist of β_1_AR and β_2_AR, reserving the catechol moiety and forming a hydrogen bond with N6.55 [54]. Therefore, there may be other receptor–ligand interactions affecting ligand efficacy to be discovered. NMR experiments showed that in β_2_AR, the chemical shift of M2.53 depends on the efficacy of the ligand [97], and dobutamine’s direct contact with TM2 (at 2.61, 2.64, and 2.65). Whether this is the basis of partial agonism of dobutamine remains to be investigated.

### 4.4. β_1_/β_2_ Subtype Selectivity

β_1_AR and β_2_AR are highly similar in sequence (identity 57% at full-length and 70% at pocket residues) and structure (root mean square deviation 0.71 Å for inactive state and 1.00 Å for G_s_-coupled active state). Despite their high homology, numerous agonists and antagonists selective for β_1_AR or β_2_AR have been discovered. Even endogenous agonist norepinephrine, small in size, thus all the interacting residues are the same in the two receptors, is approximately 10-fold selective for β_1_AR (while the other endogenous ligand epinephrine, with only one extra methyl group, is nonselective). Revealing of the molecular mechanisms of β_1_/β_2_ subtype selectivity began only recently.

In β_1_AR and β_2_AR, although all the residues at the orthosteric site are the same, some residues at the edge of the pocket are different. Large ligands may directly interact with these residues. For example, salmeterol (>1000-fold selective for β_2_AR) has a long aryloxyalkyl tail, forming additional interactions to TM2, TM3, TM7, ECL2, and ECL3. Three of the interacting residues, while mutated to the corresponding residue in β_1_AR (H6.58N, K7.32D, and Y7.35F), largely reduce the affinity of salmeterol [70]. Residues at ECL2 and the extracellular end of TM6/7 sculpt the different shapes of pocket entrances in the two β adrenergic receptors: in β_2_AR, K7.32 forms a salt bridge to D45.51, covering the space among TM2/3/7; and in human β_1_AR, a different salt bridge is formed by R6.62 to E205 (ECL2), covering the space among TM3/5/6. Therefore, ligands potentially take different paths to enter β_1_AR and β_2_AR. The passage in β_1_AR is continuously negatively charged, thus preferred by norepinephrine, while epinephrine, a secondary amine, is less affected. This hypothesis explains the fast association rate of norepinephrine in β_1_AR and is supported by free energy profiles, calculated using metadynamics simulations [43].

## 5. Mechanisms of α_2_ Receptors

### 5.1. Ligand Binding

Structures of α_2A_AR [98] and α_2C_AR [99] in an inactive state and α_2B_AR [52] in an active state are disclosed. The binding of both agonist and antagonist in α_2_ adrenergic receptors involves more π-π or cation-π interactions and fewer hydrogen bonds (Figure 6a,b), comparing to βARs (Figure 3b). The expansion of π interaction network is mainly caused by F7.39, conserved in all α_1_ARs and α_2_ARs but is N7.39 in βARs. In the agonist-bound structures, F7.39 acts as a ‘lid’, covering the imidazole ring of ligand from the extracellular side (Figure 6c). Typical α agonists and some α antagonists contain positively charged planar groups, such as imidazole, imidazoline, and guanidine [100], which fit well in this closed pocket. In inactive structures, the ‘lid’ F7.39 is pushed up to another rotamer by antagonist (Figure 6d), making the pocket larger, thus able to accommodate bulky ligands. Some α ligands with such chemical features are in clinical use, including piperazine derivatives as α_1_ antagonists (e.g., prazosin) and indole alkaloids with multiple fused rings as α_2_ antagonists (e.g., yohimbine).

### 5.2. α/β Selectivity

The binding of α agonists and antagonists relies on π interactions involving F7.39, thus, is not favored in βARs with N7.39. β ligands, instead, largely prefer to retain ethanolamine moiety [101] to form dual hydrogen bonds to N7.39 (Figure 6c). Another feature of β ligands is a single large substituent on the amino N atom. A large group at this position is not favored in αARs as it is too large to fit in the closed pocket and too flexible to trigger the ‘switching lid’. Only one substitution is allowed because only primary and secondary amines can form a hydrogen bond to N7.39 in βARs. This is validated by the polypharmacology profile of ergotamine, a tertiary amine: the binding affinities are 0.3–3 nM in α_2_ARs, 10–30 nM in α_1_ARs, and ~100 nM in β_2_AR and >10,000 nM in β_1_AR/β_3_AR [23].

### 5.3. Alternative Activation Mechanism

Activation of β_1_AR/β_2_AR requires an agonist forming a hydrogen bond to S5.46. In α_2_ARs, the catecholamine agonists can form this hydrogen bond, but many α_2_ agonists, including dexmedetomidine and clonidine, do not have polar groups at this position. Mutation S5.42A or S5.46A in α_2A_AR reduce the activation of epinephrine, but not of dexmedetomidine and clonidine [102]. Structures show that dexmedetomidine activates α_2B_ by forming π-π interactions to W6.48 and pushing it downward (Figure 6d). The downward movement of W6.48 is part of the common activation mechanism in class A GPCRs (Figure 4a). To effectively push W6.48, the ligand must compactly fill the pocket. This mechanism explains the functions of imidazole/imidazoline derivatives as α_2_ drugs: dexmedetomidine and clonidine are agonists for they fill the pocket, while tolazoline, without substituents on the benzen ring, thus being smaller in size, is an antagonist (Figure 6e). This series of imidazole/imidazoline derivatives is a special case in GPCR ligands, as normally, antagonists are larger than agonists because the pocket of inactive state is larger.

### 5.4. Pathway Selectivity

Position 6.55 is the only one in the orthosteric site that each major type of adrenergic receptor has a residue with a specific property: α_1_ARs, hydrophobic M6.55 (α_1A_AR) or L6.55 (α_1B_AR/α_1D_AR); α_2_ARs, aromatic Y6.55; βARs, polar N6.55. Interestingly, Y6.55 in α_2_ARs shows a distinct impact on different downstream pathways: α_2_ARs mainly functions through the G_i_ pathway, but they also couple to G_s_. Mutation Y6.55N demolishes G_s_ signaling but retains G_i_ signaling in α_2A_AR, suggesting that this residue has a subtle impact on receptor conformation [98]. In both active and inactive structures of α_2_ARs, Y6.55 lies horizontally, and forms van der Waals interactions with ligands from the extracellular site. In active state α_2B_AR, Y6.55 forms a hydrogen bond with S5.42 (Figure 6a). This hydrogen bond may be comparable to the hydrogen bond between N6.55 and S5.43 in βARs, while N6.55 and S5.43 affect the efficacy of agonists [70]. Structures also exhibit other possible roles of Y6.55: first, Y6.55 defines the upper boundary of the pocket (Figure 6e), thus rejecting many β ligands with large groups at the catechol recognizing side (e.g., formoterol and carvedilol); second, Y6.55 blocks the TM3/5/6 space, leaving TM2/3/7 the only passage for ligands. This might be the basis for both epinephrine and norepinephrine having similar affinities in all three α_2_ adrenergic receptors [13,43].

### 5.5. α_2A_/α_2C_ Subtype Selectivity

α_2_ARs are highly similar in sequence, but there are still α_2_ ligands with high subtype selectivity. The molecular docking of two α_2C_ antagonists JP1302 (100 folds selectivity to α_2A_AR, 50 folds to α_2B_AR [103]) and OPC-28326 (300 folds to α_2A_AR, 50 folds to α_2B_AR [104]) suggested that the two compounds, large and long, interact with non-conserved extracellular residues. Mutagenesis showed that an interaction network among Y6.58 (unique to α_2C_AR), R6.62, and D45.54 are key to the functions of JP1302 and OPC-28326, while in α_2A_AR, mutations of a different interaction network involving R7.32 (unique to α_2A_AR) restore their functions [99]. Such impact of extracellular residues on subtype selectivity is comparable to the β_1_/β_2_ selectivity case.

## 6. Allosteric Modulations of β_2_ Adrenergic Receptor

In a GPCR, a ligand may bind outside the orthosteric site and stabilize a specific state, and this phenomenon is called allosteric modulation [105,106]. An allosteric ligand increasing the signaling of orthosteric agonists is a positive allosteric modulator (PAM), and one decreasing the signaling is a negative allosteric modulator (NAM). Allosteric modulators have raised attention in drug discovery against GPCRs because targeting less-conserved allosteric sites may produce subtype selectivity. The most effective way to identify an allosteric site is determining the structure of the GPCR in complex with the allosteric modulator [107]. Structures of β_2_AR captured small molecules binding at different sites and exhibited how they affect the signaling (Figure 7). The cholesterol binding sites were thoroughly reviewed recently [108], thus is not discussed here.

### 6.1. PAM and NAM at Lipid Interface of TM3/4/5

The arrangement of seven TM helices leads to a groove at the lipid interface formed by TM3/4/5 in all classes of GPCRs. In β_2_AR, both PAM and NAM binds in this groove are identified by crystal structures (Figure 7b). The PAM and the NAM locate at different heights and stabilize different states through specific interactions: the PAM Cmpd-6FA binds close to the cytoplasmic end and interacts with ICL2 and adjacent residues on TM3 and TM4 [110]. At this region, GPCRs interact with multiple segments of Gα, and Cmpd-6FA stabilizes conformation of the receptor. The NAM AS408 binds close to the center of the receptor and interacts with TM3 and TM5 [111]. AS408 forms a hydrogen bond to the main-chain O atom of V5.45, thus stabilizing the inactive conformation of TM5 and the P-I-F microswitch motif. Both the PAM and the NAM are selective for β_2_AR over β_1_AR; for positive modulating, two key residues (F3.52/L3.52 and K4.41/R4.41 in β_2_AR/β_1_AR) are identified. Since functions of ICL2 and TM5 are common in activation process of class A GPCRs, it is not surprising that allosteric modulators binding at the same site were also discovered in other receptors: PAMs AP8 [112] and Cpd-1 [113] of free fatty acid receptor 1, PAM LY3154207 [114] of dopamine receptor D_1_, and NAMs NDT9513727 [115,116] and avancopan [117] of C5a anaphylatoxin chemotactic receptor 1.

### 6.2. NAM at Cytoplasmic Surface

GPCRs use the cytoplasmic surface to interact with intracellular partners, including G proteins and β-arrestins. Small molecules binding at this site were mainly discovered in chemokine receptors [116,117,118,119], while Cmpd-15PA in β_2_AR [120] is the only case outside the chemokine subfamily (Figure 7c). All these small molecules are NAMs. Cmpd-15PA has little collisions with G protein but stabilizes the inactive state of the receptor through extensive interactions with TM1, TM2, TM6, TM7, H8, and ICL1. Most notably, the hydrogen bond to T6.36 and π-π interaction to Y7.53 block the outward movement of TM6 and rearrangement of TM7. Although β_2_AR and β_1_AR only differ in one position (F1.60 in β_2_AR and T1.60 in β_1_AR) among all residues interacting with Cmpd-15PA, the Cmpd-15 (parent compound of Cmpd-15PA with comparable effects) decreases the EC_50_ of orthosteric agonist in only β_2_AR but not β_1_AR [121].

## 7. Structure-Based Ligand Discovery of Adrenergic Receptors

### 7.1. Virtual Screening with Molecular Docking

Structure-based virtual screening can identify ligands with novel scaffolds and chemical types [122,123]. Since structures of β_2_AR are reported, new antagonists and agonists were discovered through virtual screening with molecular docking. Docking screening identified a new antagonist with benzhydrylpiperazine moiety having high sub-nM affinity (*K*_i_ = 0.311 nM) [124]. Screening of the lead-like subset of the ZINC database [125] identified 5 hit inverse agonists (among 25 tested compounds) with *K*_i_ < 40 μM, while the strongest had *K*_i_ = 9 nM [126]. Using an active state structure, docking screening of the lead-like and fragment-like subsets of the ZINC database identified 6 agonists (4 full and 2 partial, among 22 tested compounds) [88].

### 7.2. Rational Design

Structures can guide modification of known ligands to achieve new function. Two designed β agonists, analogs of BI-167107 and epinephrine but that can form disulfide bond, covalently bind to β_2_AR with designed mutation H2.64C. They were obtained for the crystal structure in the agonist-bound inactive state [127] and nanobody-coupled active state [128], respectively. Adding of a methyl group to cyanopindolol (partial agonist of β_1_AR and β_2_AR) at a position close to S5.46 further reduced its efficacy [77]. Inspired by structures of β adrenergic receptors, five compounds based on atypical antipsychotic drug aripiprazole were designed and shown to have different effect on dopamine receptor D_2_ (one unbiased agonist, two arrestin-biased agonists, and two antagonists) [129].

### 7.3. Predicting Actions of Ligands

Molecular docking aims to identify compounds with high affinity but does not distinguish between an agonist and antagonist. Molecular interaction fingerprints derived from structures of a GPCR with various ligands have been developed to predict the actions of ligand. When only the first inactive β_2_AR structure was available, the molecular interaction fingerprint was generated based on docking poses of ligands [130]. The newest version of fingerprint was developed based on 31 structures of β_1_AR and β_2_AR [131]. Molecular docking followed by molecular interaction fingerprint (derived from 13 structures) screening identified 26 hit agonists (among 63 tested compounds) of β_2_AR with EC_50_ < 100 μM, while the strongest had EC_50_ 38 nM [132].

### 7.4. Accelerated Structure Determination

Structure-based drug design against GPCRs relies on a large number of solved structures binding various ligands. Since 2018, many structures of active state GPCRs in functional complexes have been solved with the rapidly developed cryogenic electron microscopy (cryo-EM) technique. A remarkable case is dopamine receptor D_1_: no structure was reported before February 2021, and then 11 structures were released within two months [113,133,134,135]. For adrenergic receptors, up to April 2021, a total of eight cryo-EM structures were published: one is arrestin-coupled β_1_AR and the others are G protein-coupled (four β_2_AR, G_s_ [8,68,69]; one β_1_AR, G_s_ [66]; two α_2B_AR, G_i_ and G_o_ [52]).

Crystallography is still powerful for the structural study of adrenergic receptors. Thermostabilized turkey β_1_AR can co-crystallize with small, fragment-like ligands [89]. Crystal structures of human β_1_AR were first reported in 2020, in both a nanobody-coupled active state and inactive state, exhibiting a detailed difference from turkey β_1_AR [43]. For β_2_AR, structures in complex with eight ligands were obtained using a newly developed transient ligand exchange method termed Complex-LCP (Crystallization of Membrane Proteins using transient Ligand EXchange in LCP). The Complex-LCP method relies on the serial femtosecond crystallography (SFX) data-collection approach, using extremely bright and short pulses generated by an X-ray free-electron laser (XFEL), thus collects data from small crystals at room temperature [75].

## 8. Conclusions

Structures of adrenergic receptors exhibited mechanisms of ligand-induced effects, including agonism/antagonism, partial agonism, and biased signaling. A preliminary picture of ligand function in β adrenergic receptors was generated. The picture in α_2_ is still incomplete, while that in α_1_ remains to be uncovered. Interpreting of these mechanisms would help to discover new drugs with desired effects against adrenergic receptors.

## Figures and Tables

**Figure 1 biomolecules-11-00936-f001:**
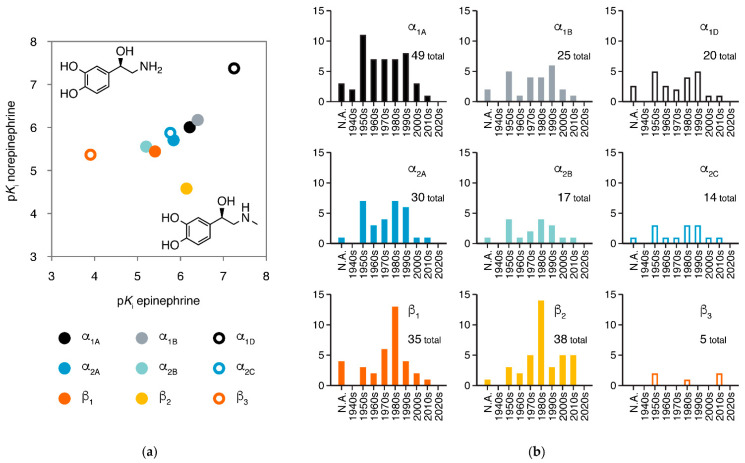
Adrenergic receptors as drug targets. (**a**) Binding affinities [12,13,14] of endogenous agonists in the nine adrenergic receptors. (**b**) Number of new drugs (approved or first marketing) for the nine adrenergic receptors in each decade. Data retrieved from Guide to PHARMACOLOGY [15] (www.guidetopharmacology.org, accessed on 25 April 2021), GPCRdb [16] (gpcrdb.org, accessed on 25 April 2021), and DrugBank [17] (go.drugbank.com, accessed on 25 April 2021).

**Figure 2 biomolecules-11-00936-f002:**
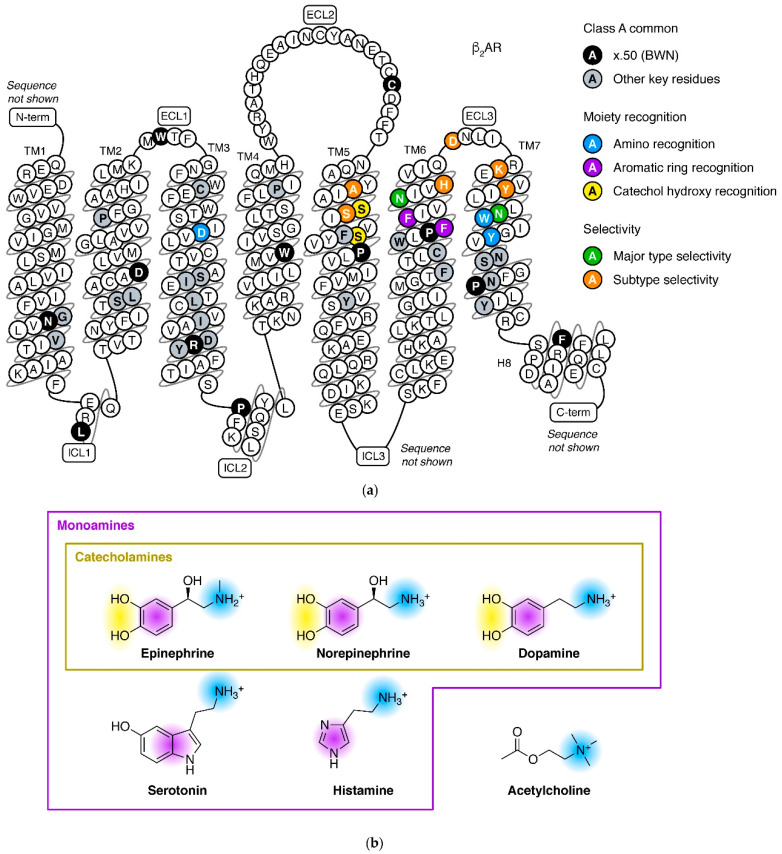
Mechanisms of ligand recognition by adrenergic receptors. (**a**) Key residues for protein structure maintaining, receptor activation, and endogenous ligand recognition, marked with color on snake plot of β_2_AR. x.50 are Ballesteros–Weinstein numbering (BWN) [22] of the most conserved residues in each segment. Sequence of N-terminus, ICL3, and C-terminus are not shown because these regions do not contain key residues and are very long. (**b**) Chemical structures of monoamine neurotransmitters (shown in protonated state) and acetylcholine. Moieties colored in consistent with interacting residues in panel (**a**).

**Figure 3 biomolecules-11-00936-f003:**
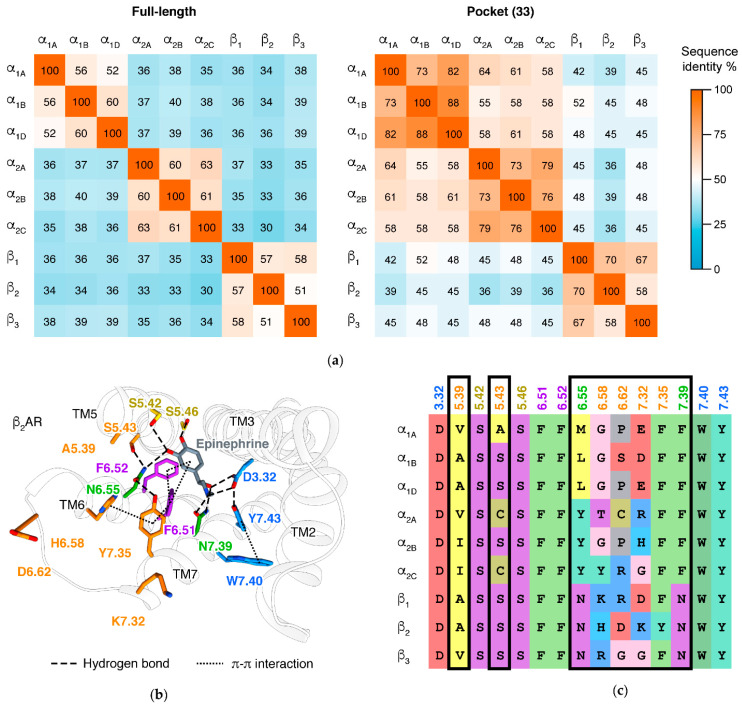
Comparison of adrenergic receptor major types and subtypes. (**a**) Sequence identity matrices of adrenergic receptors at full-length sequence (based on multiple sequence alignment generated with MAFFT v7.299b [30]) or pocket residues (33 residues: 2.61,64–65; 3.28–29,32–33,36–37; 45.50–54; 5.38–39,42–43,46; 6.48,51–52,55,58–59,62; 7.30,32,35–36,39–40,43). ‘Pocket’ is the cavity at the extracellular side of a GPCR. (**b**) Key interactions of endogenous ligand epinephrine in orthosteric site of β_2_AR/epinephrine (PDB: 4LDO [31], coloring scheme the same to Figure 2a). (**c**) Alignment of key residues for moiety recognition or type selectivity (residue BWN colored in the same scheme to Figure 2a and Figure 3b) in all the adrenergic receptors members (generated from GPCRdb plot [16], residues colored for property: red, negatively charged; blue, positively charged; yellow, hydrophobic; blue, aromatic; magenta, hydrophilic; pink, glycine; gray, proline).

**Figure 4 biomolecules-11-00936-f004:**
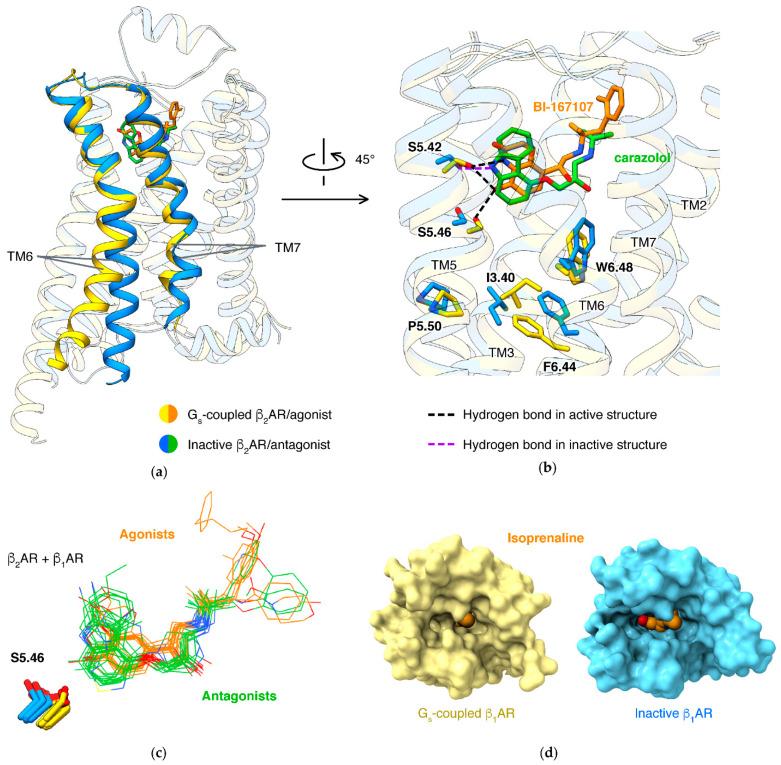
Mechanism of agonism in β adrenergic receptors. (**a**) Structures of β_2_AR in active/inactive states (PDB IDs: 3SN6 [8], 2RH1 [6]). (**b**) Interactions of agonist/antagonist to S5.42/S5.46 and microswitch of P5.50-I3.40-F6.44 motif. (**c**) Binding poses of agonists/antagonists in β_1_AR/β_2_AR [6,8,11,31,43,54,66,67,68,69,70,71,72,73,74,75,76,77]. Structures were aligned based on Cα atoms of 7 residues: 3.32; 5.42,46; 6.51–52; 7.40,43. (**d**) Binding poses of isoproterenol in β_1_ active/inactive states (PDB IDs: 7JJO [66], 2Y03 [67]).

**Figure 5 biomolecules-11-00936-f005:**
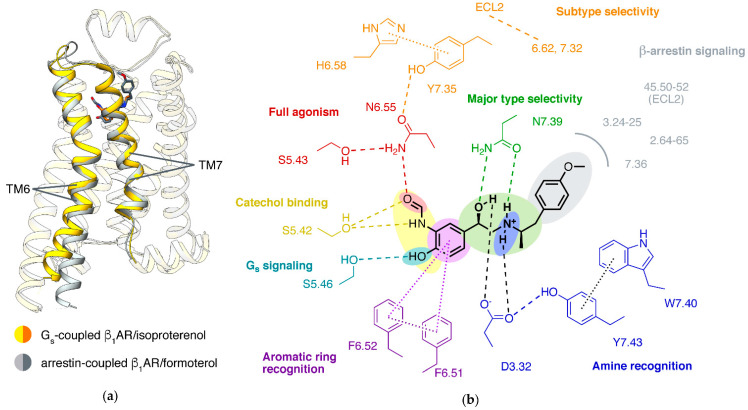
Mechanisms of effects of ligands on β adrenergic receptors. (**a**) Structures of β_1_AR coupled to G_s_ or arrestin (PDB IDs: 7JJO [66], 6TKO [11]) (**b**) Sketch map of key ligand interactions in βARs. Formoterol in β_2_AR was used for the drawing.

**Figure 6 biomolecules-11-00936-f006:**
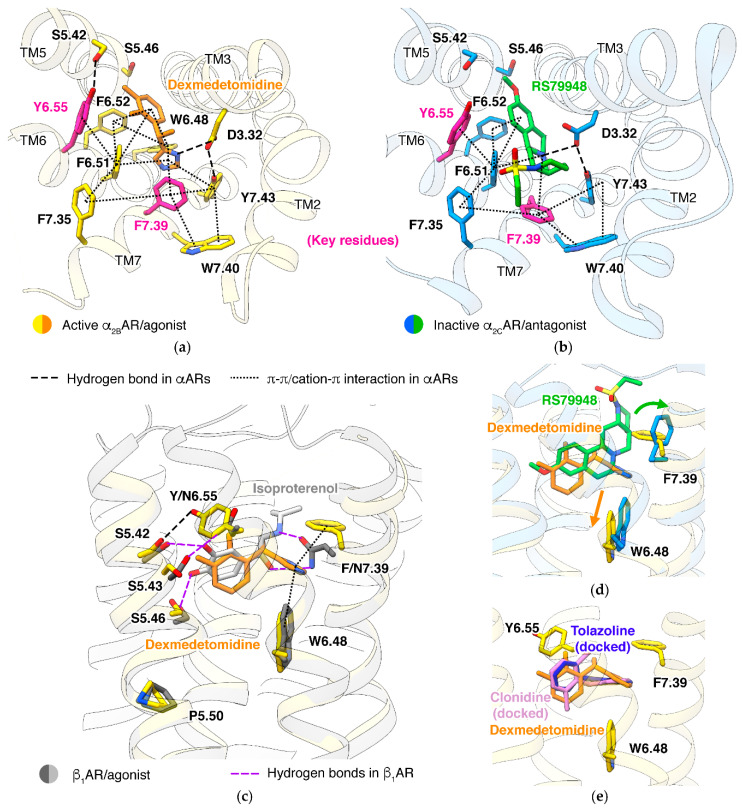
Mechanisms in α_2_ adrenergic receptors. (**a**) Agonist recognition (dexmedetomidine in α_2B_AR, PDB ID: 6K41 [52]). (**b**) Antagonist recognition (RS 79948 in α_2C_AR, PDB ID: 6KUW [99]). (**c**) Comparison of agonist recognition in α_2_ARs and βARs (isoproterenol in β_1_AR, PDB ID: 6H7J [54]). (**d**) Conformational changes in active/inactive α_2_ARs. (**e**) Binding poses of agonist clonidine and antagonist torazoline, predicted by molecular docking (Glide program in Schrodinger platform 2019-2 [38]).

**Figure 7 biomolecules-11-00936-f007:**
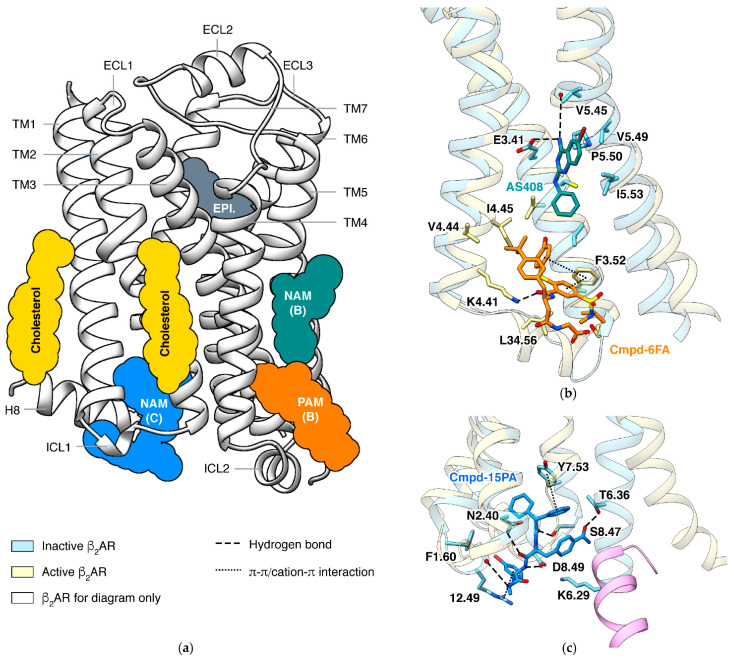
Allosteric modulating of β_2_AR. (**a**) Positions of allosteric sites (including cholesterol binding sites) identified in structures. The orthosteric site is marked by endogenous agonist epinephrine (EPI.). Positions of cholesterol are from a crystal structure in an inactive state (PDB ID: 2RH1 [6]). (**b**) PAM Cmpd-6FA and NAM AS408 in lipid interface at TM3/4/5 (PDB IDs: 6N48 [106] and 6OBA [107]). (**c**) Cytoplasmic NAM Cmpd-15PA compared to active state β_2_AR (PDB IDs: 5X7D [109] and 3SN6 [8]).

**Table 1 biomolecules-11-00936-t001:** Major types of adrenergic receptors.

Major Type	Primary Pathway	Main Effects	Indications of Agonists	Indications of Antagonists
α_1_	G_q/11_	smooth muscle contraction	vasodilatory shock, hypotension [46]	hypertension, benign prostatic hyperplasia [47]
α_2_	G_i/o_	inhibition of norepinephrine release	hypertension, pain and panic disorders [48]	erectile dysfunction [49], depression [50]
β	G_s_	β_1_: cardiac stimulation β_2_: bronchodilation β_3_: increased lipolysis	cardiogenic shock, heart failure, asthma, overactive bladder [4]	heart failure, arrhythmias, hypertension [51]

## Data Availability

Not applicable.

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
