# Peer review of "Ligands of Adrenergic Receptors: A Structural Point of View"

_biomolecules, 2021, doi:10.3390/biom11070936_

Round 1
Reviewer 1 Report
The authors provide a comprehensive overview of the structural studies of adrenergic GPCRs, covering important topics such as efficacy, bias, selectivity and more. The figures are very nice and comprehensible. Most paragraphs are sufficiently referenced. The manuscript needs editing for language, in its current form some parts are difficult to read.
Further minor points to address:
-
Please introduce the term “orthosteric binding pocket” before its first use in l. 146. I suggest with Figures 2 or 3b.
- 3.
l.98 either among the most studied or well-studied - 4.3.
- please introduce the term ligand efficacy
- "less movement" - clarify "less", it does not refer to a smaller TM6 movement but to a smaller shift of the conformational equilibrium.
- in this context the DEER studies by Manglik et al. Cell, 2015 and Lerch et al. PNAS, 2020 should be referenced, as they directly measure population shifts of TM6 movement - 4.4
- l. 211 Please use: Ligands potentially/probably/likely take different paths to enter b1AR and b2AR. or Simulations suggest… - 5.3
- please provide reference(s) for the alternative activation mechanism. - 5.4
- l. 266 please correct Y7.55N to Y6.55N
- ll. 270-277 Please provide reference(s) for this interesting mechanism - Figure 7: Allosteric modulation
- 6.2
- l. 332 explain the term “potency” - 7.
major language editing necessary - 7.1
- add equal signs “=” to Ki values - 7.2
- l. 353 what is a balanced agonist, please explain - The term “complex structure” is colloquial. It should be replaced with structure of the complex (of X and Y), or similar.
Author Response
The authors provide a comprehensive overview of the structural studies of adrenergic GPCRs, covering important topics such as efficacy, bias, selectivity and more. The figures are very nice and comprehensible. Most paragraphs are sufficiently referenced. The manuscript needs editing for language, in its current form some parts are difficult to read.
Answer: Thank you for the suggestions. We have edited our text to increase readability.
Further minor points to address:
Please introduce the term “orthosteric binding pocket” before its first use in l. 146. I suggest with Figures 2 or 3b.
Answer: We added introduction to orthosteric site in l. 59. We also modified captions of Figure 3b to clarify we were describing the orthosteric site (l. 94).
3.
l.98 either among the most studied or well-studied
Answer: We have modified the text to “most studied” (l. 99).
4.3.
- please introduce the term ligand efficacy
Answer: We have added introduction to the term ligand efficacy (l. 180-181).
- "less movement" - clarify "less", it does not refer to a smaller TM6 movement but to a smaller shift of the conformational equilibrium.
Answer: We modified “less movement” to “a smaller shift of conformational equilibrium” (l. 179).
- in this context the DEER studies by Manglik et al. Cell, 2015 and Lerch et al. PNAS, 2020 should be referenced, as they directly measure population shifts of TM6 movement
Answer: We have added these references (l. 177-178).
4.4
- l. 211 Please use: Ligands potentially/probably/likely take different paths to enter b1AR and b2AR. or Simulations suggest…
Answer: We have modified the sentence to “ligands potentially take different paths to enter…” (l. 213).
5.3
- please provide reference(s) for the alternative activation mechanism.
Answer: We added results from literature that mutations S5.46A/S5.42 A have different effects on different ligand types and cited the literature (l. 258-259).
5.4
- l. 266 please correct Y7.55N to Y6.55N
Answer: We corrected the numbering (l. 273).
- ll. 270-277 Please provide reference(s) for this interesting mechanism
Answer: We added corresponding references (l. 279, 284).
Figure 7: Allosteric modulation
6.2
- l. 332 explain the term “potency”
Answer: We used the term “EC50” from literature instead (l. 340).
7.
major language editing necessary
Answer: We have edited our text to increase readability.
7.1
- add equal signs “=” to Ki values
Answer: We have added the equal signs following Ki values (l. 348, 351).
7.2
- l. 353 what is a balanced agonist, please explain
Answer: We used the precise term 'unbiased agonist' instead (l. 361).
The term “complex structure” is colloquial. It should be replaced with structure of the complex (of X and Y), or similar.
Answer: We have modified this phrase to “structure of … in complex with…” (l. 160, 302) or simply “structure” (l. 366, 380)
Reviewer 2 Report
This is a timely review describing the structural basis of ligand binding by adrenergic receptors. Subtype specific differences between the major classes of adrenergic receptors are described with respect to ligand binding preference. The features of agonists, antagonists, and allosteric modulators that modulate receptor conformation are also discussed. The review would be improved through changes that increase readability and context.
What is meant by the statement "The three major types of adrenergic receptors are not neighbors in the tree of aminergic receptors"? As no tree is shown, its hard to appreciate the point of this sentence.
Figure 2a is confusing. What is denoted by the residues in black circles, which are labelled as x.50? The terms describing the highlighted residues are not clear, for example "other key residues", "major type" and "subtype". In the caption, segments that are showed in dashed lines are stated as being hidden. Does this mean disordered in published structures?
In Figure 3a, the pocket residues are shown, but its not clear what the pocket is referring to. Is it the orthosteric site of the receptors? The term "pocket" is used throughout the review, but it is never defined.
Figure 3c shows a partial sequence alignment of conserved positions across receptors. Do the selected residues correspond to the ligand binding residues across receptors? The caption needs more information to explain why some of the positions are boxed, and how they are colored by property.
It would be helpful to show the structural changes that occur upon beta-arresting binding to the receptor, similar to Figure 4, which shows the changes upon G protein binding.
In Figure 5, the b-arrestin and subtype selectivity regions shown are very vague.
In Figure 6, it would be helpful to show a comparison between the alpha-AR and beta-AR orthosteric sites, rather than relying solely on a description. Also, what does the color coding in panels a and b corespond to?
In lines 225-227, it seems highly unlikely that the imidizole, imdazoline, and guanidine groups are negatively charged
In line 331, what is the <4.0Å in reference to? how Cmpd-15PA binds? The difference between F1.60 and T1.60 in the b2 and b1-ARs?
Author Response
This is a timely review describing the structural basis of ligand binding by adrenergic receptors. Subtype specific differences between the major classes of adrenergic receptors are described with respect to ligand binding preference. The features of agonists, antagonists, and allosteric modulators that modulate receptor conformation are also discussed. The review would be improved through changes that increase readability and context.
Answer: Thank you for the suggestions. We have edited our text to increase readability. We also added introductions to terms: orthosteric site (l. 59) and ligand efficacy (l. 180-181),
What is meant by the statement "The three major types of adrenergic receptors are not neighbors in the tree of aminergic receptors"? As no tree is shown, its hard to appreciate the point of this sentence.
Answer: We found this statement is not related to the topic and deleted the two sentences (l. 62).
Figure 2a is confusing. What is denoted by the residues in black circles, which are labelled as x.50? The terms describing the highlighted residues are not clear, for example "other key residues", "major type" and "subtype". In the caption, segments that are showed in dashed lines are stated as being hidden. Does this mean disordered in published structures?
Answer: 'x.50' are Ballesteros-Weinstein numbers of the most conserved residues in each helix or loop segment. ‘Other key residues’ are key residues beside x.50. We added note ‘BWN’ in the figure (l. 83) and the explanation in figure caption (l. 85-86).
We modified terms of moiety recognition residues and type selectivity in the figure to make them clear.
Sequence of segments in dashed lines were not shown because they do not contain key residues and are very long. We replaced the dashed lines for the confusing to disordered region, and added note 'sequence not shown' in the figure. We added reason for hide the sequences in caption (l. 86-87).
In Figure 3a, the pocket residues are shown, but its not clear what the pocket is referring to. Is it the orthosteric site of the receptors? The term "pocket" is used throughout the review, but it is never defined.
Answer: We added note that it is orthosteric site in the caption of Figure 3a (l. 94).
We defined the term ‘pocket’ in caption of Figure 3a (l. 93).
Figure 3c shows a partial sequence alignment of conserved positions across receptors. Do the selected residues correspond to the ligand binding residues across receptors? The caption needs more information to explain why some of the positions are boxed, and how they are colored by property.
Answer: Residues in Figure 3c are key residues for recognizing endogenous or deciding type selectivity. Residue number are colored in the same scheme as Figure 2a and 3b. Residues colored for property: red, negatively charged; blue, positively charged; yellow, hydrophobic; blue, aromatic; magenta, hydrophilic; pink, glycine; gray, proline. We added these notes in caption of Figure 3a (l. 95-97).
It would be helpful to show the structural changes that occur upon beta-arresting binding to the receptor, similar to Figure 4, which shows the changes upon G protein binding.
Answer: We added a subplot Figure 5a to show the receptor bound to beta-arrestin (Figure 5a, l. 218).
In Figure 5, the b-arrestin and subtype selectivity regions shown are very vague.
Answer: We specified the regions that involved in the interactions on these segments (Figure 5b, l. 218).
In Figure 6, it would be helpful to show a comparison between the alpha-AR and beta-AR orthosteric sites, rather than relying solely on a description. Also, what does the color coding in panels a and b corespond to?
Answer: We added a figure to compare orthosteric sites of alpha-AR and beta-AR (Figure 6c, l. 237). Figure 6c in the original version was renumbered as 6d in the revised version, and 6d in the original version was removed.
In lines 225-227, it seems highly unlikely that the imidizole, imdazoline, and guanidine groups are negatively charged.
Answer: We corrected the phrase to "positively charged" (l. 230).
In line 331, what is the <4.0Å in reference to? how Cmpd-15PA binds? The difference between F1.60 and T1.60 in the b2 and b1-ARs?
Answer: We deleted the mistakenly used “T1.60”. We deleted the confusing “<4.0Å” and modified the phrase to “among all residues interacting with Cmpd-15PA”. (l. 339).